# Robust, automated sleep scoring by a compact neural network with distributional shift correction

Zeke Barger[1]*, Charles G. Frye[1,2], Danqian Liu[3], Yang Dan[1,3], Kristofer E. Bouchard[1,2,4]

**1** Helen Wills Neuroscience Institute, University of California, Berkeley, California, United States of America, **2** Redwood Center for Theoretical Neuroscience, University of California, Berkeley, California, United States of America, **3** Department of Molecular and Cellular Biology, Howard Hughes Medical Institute, University of California, Berkeley, California, United States of America, **4** Biological Systems and Engineering Division, Lawrence Berkeley National Laboratory, Berkeley, California, United States of America

* zeke@berkeley.edu

**Data Availability Statement:** All EEG/EMG recordings and sleep scoring data are available at https://osf.io/py5eb/ Software for our algorithm is

## Abstract

Studying the biology of sleep requires the accurate assessment of the state of experimental subjects, and manual analysis of relevant data is a major bottleneck. Recently, deep learning applied to electroencephalogram and electromyogram data has shown great promise as a sleep scoring method, approaching the limits of inter-rater reliability. As with any machine learning algorithm, the inputs to a sleep scoring classifier are typically standardized in order to remove distributional shift caused by variability in the signal collection process. However, in scientific data, experimental manipulations introduce variability that should not be removed. For example, in sleep scoring, the fraction of time spent in each arousal state can vary between control and experimental subjects. We introduce a standardization method, *mixture z-scoring*, that preserves this crucial form of distributional shift. Using both a simulated experiment and mouse *in vivo* data, we demonstrate that a common standardization method used by state-of-the-art sleep scoring algorithms introduces systematic bias, but that mixture *z*-scoring does not. We present a free, open-source user interface that uses a compact neural network and mixture *z*-scoring to allow for rapid sleep scoring with accuracy that compares well to contemporary methods. This work provides a set of computational tools for the robust automation of sleep scoring.

## Introduction

Sleep is a fundamental animal behavior and has long been the subject of intensive basic and clinical research. Mice are commonly chosen as a model organism for sleep research thanks to the wide range of genetic tools that enable manipulation of sleep-relevant neuronal ensembles and characterization of sleep phenotypes. In order to measure the effect of an experimental manipulation on the quantity or timing of sleep stages, it is necessary to score a subject's sleep stage at each point in time. In mice, each epoch is typically assigned to one of three stages based on patterns of electroencephalogram (EEG) and electromyogram (EMG) activity

available at https://github.com/zekebarger/AccuSleep.

**Funding:** ZB and CF were supported by the National Science Foundation Graduate Research Fellowship Program under Grant No. DGE 1752814 (https://www.nsfgrfp.org/). DL and YD were supported by the Howard Hughes Medical Institute (https://www.hhmi.org/). CF and KEB were supported by a Laboratory Directed Research and Development (LDRD) funding from Berkeley Lab, provided by the Director, Office of Science, of the U.S. Department of Energy under Contract No. DE-AC02-05CH11231 (https://www2.lbl.gov/DIR/LDRD/). The funders had no role in study design, data collection and analysis, decision to publish, or preparation of the manuscript.

**Competing interests:** The authors have declared that no competing interests exist.

(Fig 1): rapid eye movement (REM) sleep, with a high ratio of theta (6-8 Hz) to delta (1-4 Hz) EEG activity and low muscle tone; non-REM (NREM) sleep, with a low theta/delta ratio and low muscle tone; and wakefulness, with high muscle tone and high-frequency, low-amplitude EEG activity.

Manual inspection of the EEG and EMG signals remains the most widely used method for mouse sleep scoring. However, this process is time-intensive and therefore scales poorly with the number of subjects and recordings, motivating efforts to develop scoring methods that are partially or completely automated. Shallow decision trees [1–5], which require a user to define thresholds in a low-dimensional feature space (e.g., theta/delta ratio and EMG activity), are one such approach. However, since the three classes are not entirely separable in these low-dimensional spaces, efforts have been made to build classifiers that use machine learning to exploit a larger number of hand-tuned features [6–8]. Most recently, there have been successes in using models trained directly on EEG/EMG data without feature engineering, either in the form of spectrograms [9, 10] or unprocessed signals [11, 12]. The accuracy of these methods on held-out test sets can be close to the inter-rater reliability of expert scorers [10], suggesting that further feature or architecture engineering of EEG/EMG-based sleep scoring algorithms will yield diminishing returns.

Generalization to test datasets that differ from training datasets, however, remains a concern, both for machine learning in general [13] and for automated sleep scoring in particular. Changes in the distribution of test data, called *distributional shift*, can cause misclassification errors when items of one class in the test set artifactually resemble, due to the shift, items of another class in the training set. Simple forms of distributional shift can be solved by a standardization procedure, such as *z*-scoring. In a review of automated sleep scoring methods, Katsageorgiou et al. found that the choice of standardization procedure can be more important for classification accuracy than the choice of the classifier itself [14].

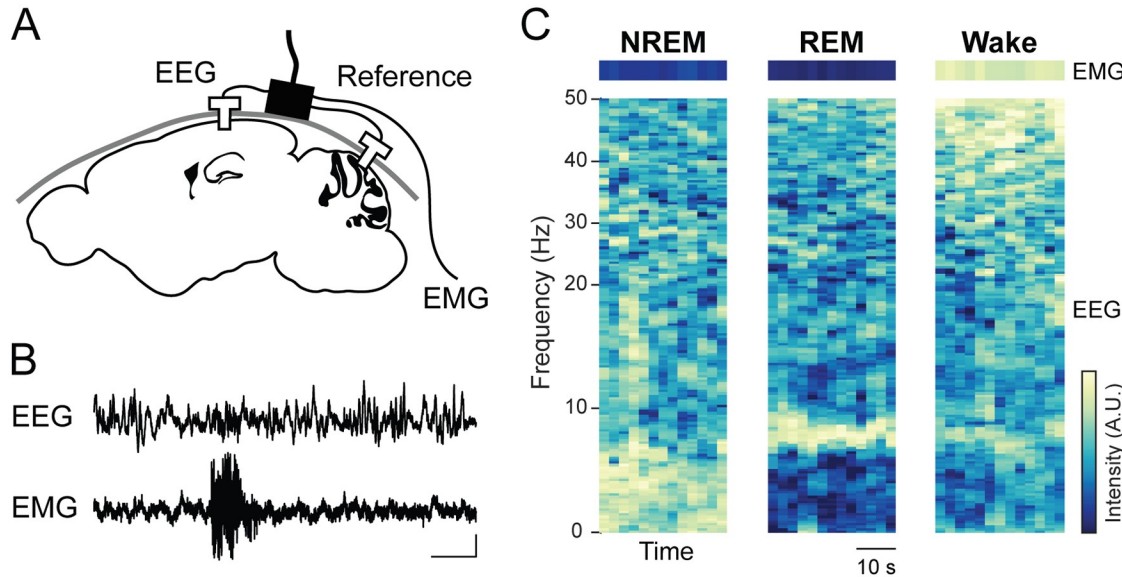

**Fig 1. Overview of the signal collection process for sleep scoring in mice.** A: schematic of EEG and EMG recordings. An EEG electrode is inserted over the hippocampus, a reference electrode is placed in the cerebellum, and an EMG electrode is inserted into the neck musculature. B: sample EEG and EMG recordings. Scale bar: 1 s, 0.25 mV. C: example EEG spectrograms and root-mean-square EMG activity for each sleep stage.

Despite the demonstrated importance of distributional shift, methods to mitigate its impact are limited [13]. We address the problem directly, focusing on two sources of distributional shift in the context of sleep scoring: *nuisance variability*, caused by changes in the way signals are recorded, and *class balance variability*, caused by changes in the time spent in each sleep stage. Both forms of distributional shift might be present in a single dataset simultaneously. However, unlike nuisance variability, class balance variability should not be removed because the primary motivation for sleep scoring is often to detect altered sleep behavior.

We use both a simple model and mouse *in vivo* data to demonstrate that standard *z*-scoring, which aims to remove nuisance variability, inappropriately reduces class balance variability. Therefore, classification algorithms using standard *z*-scoring as a preprocessing step will be biased towards underestimating changes in class balance relative to their training data, which limits their applicability in research settings where the fraction of time spent in each sleep stage is of interest. We developed a method, *mixture z-scoring*, for standardizing features of the EEG and EMG signals that disentangles nuisance and class balance variability using a small amount of labeled data for each subject. Because this requires brief but non-trivial user interaction with the recordings, we also present a free, open-source user interface that allows for rapid manual sleep scoring for standardization purposes followed by machine learning-based, automated sleep scoring. The software is available at https://github.com/zekebarger/AccuSleep.

## Results

### Mixture *z*-scoring corrects for distributional shift in simulated data

In this section, we use a simulated experiment (Fig 2) to model distributional shift in its simplest form in order to illustrate the problems caused by standard *z*-scoring and resolved by mixture *z*-scoring. Empirical findings in the next section indicate that these results generalize to a more realistic setting.

The simulation is designed as follows: two experimental subjects, Subject I and Subject II, spend different amounts of time in each of two states, state 1 and state 2. In order to detect and quantify this difference, a data feature, $\delta$, is measured from each subject and paired observations of $\delta$ and ground truth class labels from Subject I are used to train a logistic regression model. This model is then used to classify observations from Subject II. $\delta$ is drawn from a mixture of Gaussians, with one Gaussian for each class. This scenario is analogous to an experiment where Subject I is a control mouse, Subject II is a mouse undergoing sleep deprivation, state 1 is wakefulness, state 2 is sleep, and $\delta$ is the delta power in the EEG signal recorded from each mouse. The process of training the classifier on Subject I and then applying it on Subject II is analogous to the common practice of developing algorithms on wild-type populations but then applying them on wild-type and experimental subjects.

From Fig 2A, which shows kernel density estimates for the marginal and class-conditional distributions for both subjects, it can be seen that the distribution of $\delta$ for Subject I differs from that for Subject II. When the distribution of the data on which a classifier is applied differs from the training distribution, we say that there has been distributional shift. Class balance variability and nuisance variability are two major sources of distributional shift in scientific applications of classification algorithms.

Class balance variability is often the substance of the scientific inquiry that a classifier is meant to support. An experimental intervention, such as a drug, is expected to alter the amount of time spent in one or more sleep states and classification algorithms for sleep scoring are to be used to detect this change. The datasets in this simulated experiment have different class balances: Subject II spends 30% of the time in state 2 (blue), while Subject I spends 70%

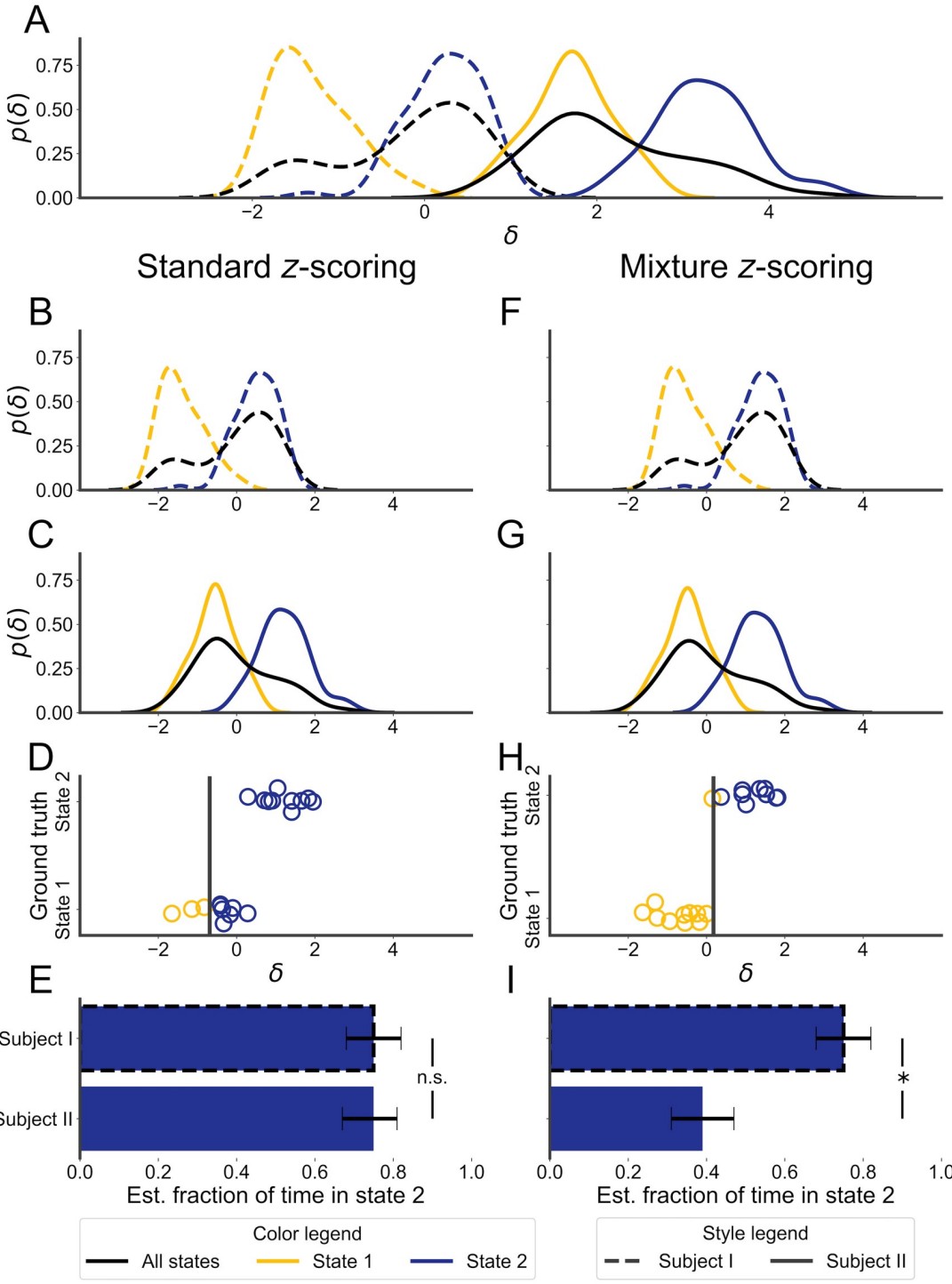

**Fig 2. Correcting for distributional shift prevents a false negative in a simple model.** A: marginal and class-conditional distributions for a feature, $\delta$, recorded from synthetic Subject I (dashed lines), which is in state 2 70% of the time, and from synthetic Subject II (solid lines), which is in state 2 30% of the time. Marginal distributions are in black, the distribution for state 2 in yellow, state 1 in blue. The distributions differ in class balance and by an affine shift. Left column, B-E: the results of applying standard $z$-scoring to this synthetic data. The $x$-axis, representing the value of $\delta$, is shared across B-D. In B and C, the marginal and class-conditional distributions are plotted after standard $z$-scoring. D: the output of a logistic classifier trained on the $z$-scored data from Subject I (in color; decision threshold represented by gray bar) compared to the ground-truth state (on the $y$-axis) for a selection of data points from Subject II. The $y$-values are jittered to improve legibility. E: the estimated fraction of time in state 2 using labels given by the classifier from D. Error bars show approximate 95% confidence intervals (CIs) for this fraction, obtained by boot-strapping. The symbol "n.s." indicates that the estimated fraction for Subject II fell within the 95% CI

for Subject I. Right column, F-I: as B-E, but using mixture *z*-scoring to correct for distributional shift. Each panel on the right-hand-side shares its y-axis with the matching panel on the left-hand side. The * symbol indicates that the estimated fraction for Subject II fell outside the 95% CI for Subject I.

of the time in that state. This difference can be seen in the marginal distributions of $\delta$ (black curves) for each subject.

In contrast, changes in the distribution of $\delta$ due to nuisance variability should be removed so that observations from different subjects can be compared. Common sources of nuisance variability in the context of sleep scoring include the use of different types of recording equipment and different implantation sites for recording electrodes. The simplest form of nuisance variability is an affine transformation. In this example, the data from Subject II have the same class-conditional distributions as in Subject I, except for an affine transformation.

For an otherwise fixed distribution, an affine transformation can be undone by means of *z*-scoring. That is, the mean, $\mu$, and standard deviation, $\sigma$, of measurements $\Phi$ are calculated for each subject and then values, $Z$, known as *z*-scores are computed:

$$Z = \frac{\Phi - \mu}{\sigma}. \tag{1}$$

If the distributions of measurements from two subjects differ only by an affine transformation, then after the application of *z*-scoring, they will be identical. Furthermore, they will have mean 0 and standard deviation 1, and so this procedure is also called *standardization*. It is typically beneficial for machine learning algorithms to operate on standardized features [15].

However, as panels B-F of Fig 2 demonstrate, when other sources of distributional shift are present, standard *z*-scoring is inappropriate. Standard *z*-scoring uses the marginal statistics of $\delta$, which are dependent on both nuisance and class balance variability. If we view the marginal distribution of $\delta$ as a mixture of the class-conditional distributions for each class, its mean and variance are functions of the mixture weights (equivalently, the marginal probabilities of each class label) and the mean and variance of each class-conditional distribution. Its mean is given by the weighted average of the conditional means, while the variance is given by the law of total variance (Eq 13, Methods).

After standard *z*-scoring, the class-conditional distributions of $\delta$ from the two subjects do not align as desired (compare the yellow curves in Fig 2B and 2C). The result is that a classifier trained to high performance on the *z*-scored data from Subject I will perform poorly on the *z*-scored data from Subject II (Fig 2D): a large fraction of observations are mislabeled as state 2 as the decision boundary aligns with the center of the distribution for state 1. In this case, the introduced bias is opposite and almost equal to the effect of the change in class balance, leading to the misclassification of a significant fraction of the observations in Subject II as state 2 when they should be state 1. The resulting bias in the estimate of the effect size of the difference leads to a reduction in power and a false negative result for a bootstrapping test (Fig 2E). Because *z*-scoring is agnostic to class label, it is unable to disentangle the effect of the class balance from the effect of nuisance variability on the marginal statistics of $\delta$.

In order to remove nuisance variability but retain class balance variability, we introduce *mixture z-scoring*, an alternative form of *z*-scoring inspired by viewing the marginal distribution of the measurements as a mixture of class-conditional distributions. The mixture *z*-scored values $Z_M$ corresponding to measured feature values $\Phi$ are computed as:

$$Z_M = \frac{\Phi - w^\top \hat{\mu}}{\sqrt{w^\top (\hat{\sigma}^2 + (\hat{\mu} - w^\top \hat{\mu})^2)}} \tag{2}$$

where $\hat{\mu}$ and $\hat{\sigma}$ are vectors of label-conditioned means and standard deviations and $w$ is a fixed vector that sums to 1. The value of the denominator and the value subtracted from $\Phi$ in the numerator are the mixture $z$-scoring parameters, by analogy with the $z$-scoring parameters. Note that conditioning on labels means that mixture $z$-scoring requires some labeled data. If $w$ is equal to the actual proportions of class labels for the values $\Phi$, then the mixture $z$-score parameters are the same as in standard $z$-scoring. When $w$ is not equal to the class balance, then the mixture $z$-score parameters are equal to what the $z$-score parameters would have been, had the class balance of $\Phi$ been $w$. Mixture $z$-scoring thus removes any effect of affine nuisance variability on the marginal distribution of $\delta$ while preserving any effect of class balance variability. We refer to this process as *mixture standardization*, by analogy with $z$-score standardization. See the Methods section for further details.

The results of using mixture $z$-scoring are shown in Fig 2, panels F-I. Panels F and G show the marginal and class-conditional distributions of mixture-standardized $\delta$ for Subjects I and II, respectively. Note that the class-conditional distributions are aligned, unlike in panels B and C. The result is that a classifier trained to high accuracy on the mixture-standardized data from Subject I also performs well on the data from Subject II when it is mixture-standardized with the same weights (panel H). A bootstrapping test comparing the time spent in state 2 for the two subjects now returns a true positive (panel I).

This example demonstrates that standard $z$-scoring of features used as inputs to a classifier can, in a simple case, lead to systematic classification errors when the class balance of the data on which the classifier is applied differs from the balance during training. This systematic error can lead to incorrect scientific conclusions. However, mixture $z$-scoring, which standardizes data without removing class balance variability, substantially reduces the error rate.

## Mixture $z$-scoring reduces bias when classifying mouse *in vivo* data

While the above section indicates that mixture $z$-scoring can correct for distributional shift in a simulated experiment with a simple classifier, it remains to be seen whether distributional shift still poses a problem in a more realistic setting with state-of-the-art methods. To address this question, we applied two automated sleep scoring algorithms to mouse EEG and EMG recordings:

1. **SPINDLE**
   This algorithm, from [10], comprises a convolutional neural network (CNN) and a hidden Markov model (HMM). The CNN operates on multi-channel EEG and EMG spectrograms and comprises max-pooling, convolution, and max-pooling followed by two fully-connected layers. It has 6.8M parameters, distributed primarily in the first fully-connected layer. The HMM is used to constrain the transitions between sleep stages predicted by the CNN. See [10] for details. EMG activity and individual frequency bands of the spectrogram are log-transformed and, importantly, $z$-scored on a per-recording basis.

2. **Sleep Scoring Artificial Neural Network (SS-ANN)**
   We implemented a simple CNN with fewer than 20K learnable parameters that, similar to SPINDLE, operates on log-transformed EEG spectrograms and EMG activity. We refer to this network as SS-ANN. It uses three convolution-ReLU-maxpool modules with batch normalization, followed by a linear classifier. See the Methods section for details of the network architecture and a list of the datasets used for training and testing SS-ANN in each experiment.

Each algorithm was applied to recordings from three wild-type mice. We applied SPINDLE to recordings from Cohort A in [10]. We applied SS-ANN to a separate set of recordings

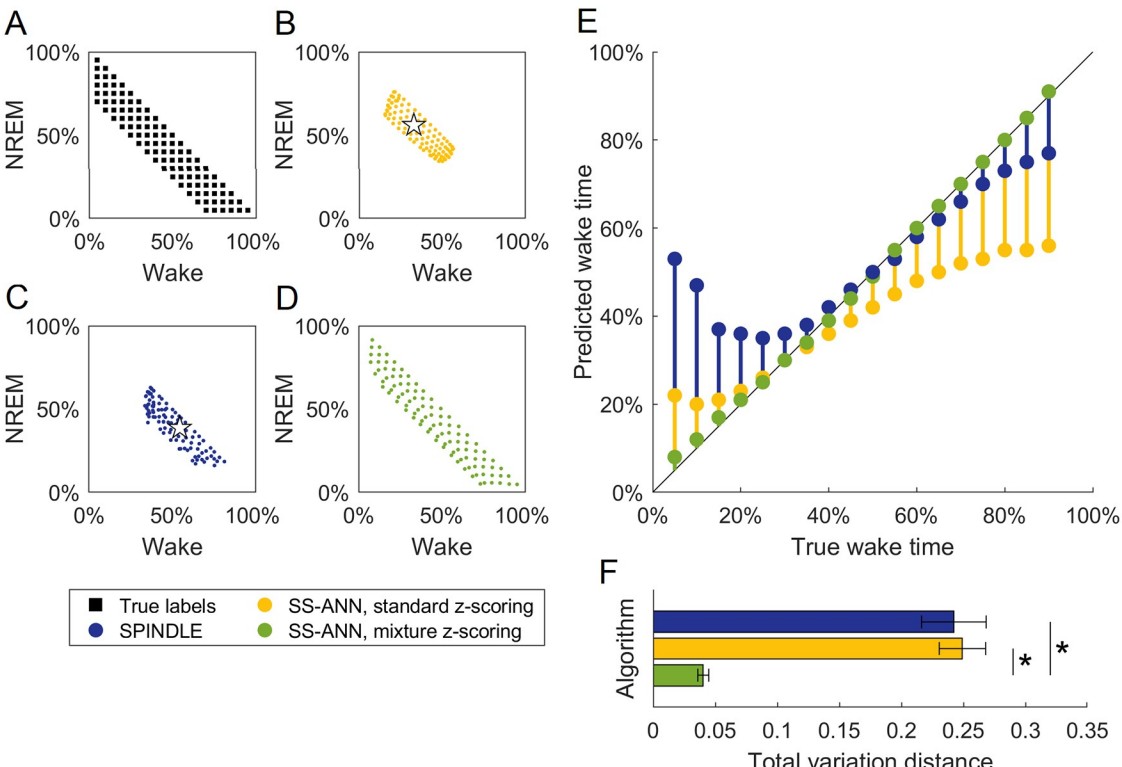

**Fig 3. Comparison of sleep scoring algorithms on recordings with programmatically varied class balances.** EEG and EMG recordings were programmatically altered to have different proportions of each sleep stage. A-D: Each marker indicates the amount of wakefulness and NREM in recordings with a given class balance, averaged across three mice (one recording per mouse). A: class balances of the recordings according to ground truth manual labels. B: predictions by SS-ANN with standard *z*-scoring for each recording. Star indicates the class balance in the training dataset for SS-ANN. C: as B, for SPINDLE. Star indicates class balance of training data for SPINDLE. D: as B, for SS-ANN with mixture *z*-scoring. E: predicted amounts of wakefulness in recordings containing 5% REM, according to each algorithm. F: mean total variation distance between class balances of algorithmic predictions and manual labels. Error bars show SEM. Asterisk indicates *p*-value < 0.001 (Student's *t*-test).

which we collected and scored (see Methods). This ensured that any observed bias in either method would not be a result of differences between the experts who scored the training datasets.

To introduce class balance variability and produce a controlled simulation of the effect of experimental manipulations, we programmatically varied the amounts of NREM sleep, REM sleep, and wakefulness in each recording (Fig 3A). Rebalancing was achieved by randomly removing bouts of each sleep stage until a specified balance was reached. To preserve the natural temporal structure of the bouts as much as possible, we included at least eight seconds of NREM before, and four seconds of wakefulness after, each REM bout. We considered class balances in the range of 5-95% wakefulness, 5-95% NREM, and 0-25% REM because class balances outside this range are unlikely to be observed in practice.

In order to demonstrate that any observed biases are general to *z*-scoring, rather than specific to SPINDLE, and that mixture *z*-scoring can eliminate them, we trained two different versions of SS-ANN. For one, the inputs during both training and testing were preprocessed with standard *z*-scoring. For the other, both inputs were preprocessed using mixture *z*-scoring with weight vector *w* given by the class balance in the training set. Mixture *z*-scoring requires a small amount of labeled data from each class for each subject, which we took from separate recordings from the three mice in this dataset.

The results of these numerical experiments are shown in Fig 3. When using standard $z$-scoring, both algorithms showed a classification bias that typically made the aggregate label distribution look more like the class balance of their training datasets (Fig 3B and 3C). The magnitude of the classification error increased as the class balance was shifted further from the balance of the training data (Fig 3E). These results indicate that this bias occurs across classifiers. However, when mixture $z$-scoring was used as a preprocessing step, there was a dramatic reduction in bias: estimated label fractions were close to the true fractions in all cases (Fig 3E), eliminating the contraction towards the class balance of the training data (Fig 3D).

We quantified the bias in estimation by measuring the total variation distance between the ground truth label fractions and the label fractions estimated by each classifier (Fig 3F). The total variation distance, $\delta$, between two distributions $q$ and $q'$ is $\delta(q, q') = \|q-q'\|^1$. We found that there was no significant difference between the two algorithms when standard $z$-scoring was used (mean for SPINDLE: 0.24, mean for SS-ANN: 0.25, Student's $t$ = -0.22, $p$ = 0.83). We further found that both SPINDLE and SS-ANN had substantially greater bias when using standard $z$-scoring than did SS-ANN using mixture $z$-scoring (mean for SS-ANN with mixture: 0.04, SPINDLE vs SS-ANN mixture $t$ = 8.19, $p \ll 0.01$; SS-ANN standard vs SS-ANN mixture $t$ = 10.33, $p \ll 0.01$). These results demonstrate that while standard $z$-scoring introduces substantial bias in state-of-the-art machine learning classifiers in cases when class balance variability is present, methods such as mixture $z$-scoring can significantly reduce this bias.

## Validation of SS-ANN

To demonstrate the utility of SS-ANN paired with mixture $z$-scoring for automated sleep scoring, we evaluated their performance using several different metrics (Fig 4). On held-out test data, we found that SS-ANN achieved 96.8% accuracy against expert annotations, comparable to the range of values reported for inter-scorer agreement (typically in the range of 90-96%) [10, 16, 17] and to the performance of SPINDLE (95-96%, see Methods), despite SS-ANN having over 300 times fewer parameters. Agreement was also high for each class individually, as evidenced by the high values on the diagonal of the confusion matrix and low values off of it (Fig 4A). The receiver operating characteristic (ROC) curves for each class were far away from the unity line (bootstrap $p$-value $\ll 0.001$) (Fig 4B).

The experiment in Fig 3 demonstrated that SS-ANN can generalize across simulated experimental conditions that create different class balances. To determine whether it can also generalize across subjects, we trained and tested the network on data from each individual subject in our cohort (Fig 4C). Classification accuracy was comparable to rates of inter-scorer agreement for all train-test pairs despite the reduced size of the training sets, indicating good generalization.

Finally, we investigated the relationship between classification accuracy and the amount of labeled data used for mixture $z$-scoring. As described in the Methods section, mixture $z$-scoring requires labeled data from each subject in order to estimate a mean and variance for each data feature within each class. Though the class balance of the labeled sample has no direct effect on the estimation of these parameters, it is important to determine how many labeled epochs are required to required to attain accurate enough parameter estimates to support classification.

We held out a number of labeled epochs chosen at random from EEG/EMG recordings and used these to perform mixture $z$-scoring followed by automatic classification of the remaining epochs. Classification accuracy increased with larger held-out portions until reaching a plateau at approximately 10 minutes of labeled data (Fig 4D). Note that performance is already high for even small sample sizes, under one minute. This should not be taken to indicate that

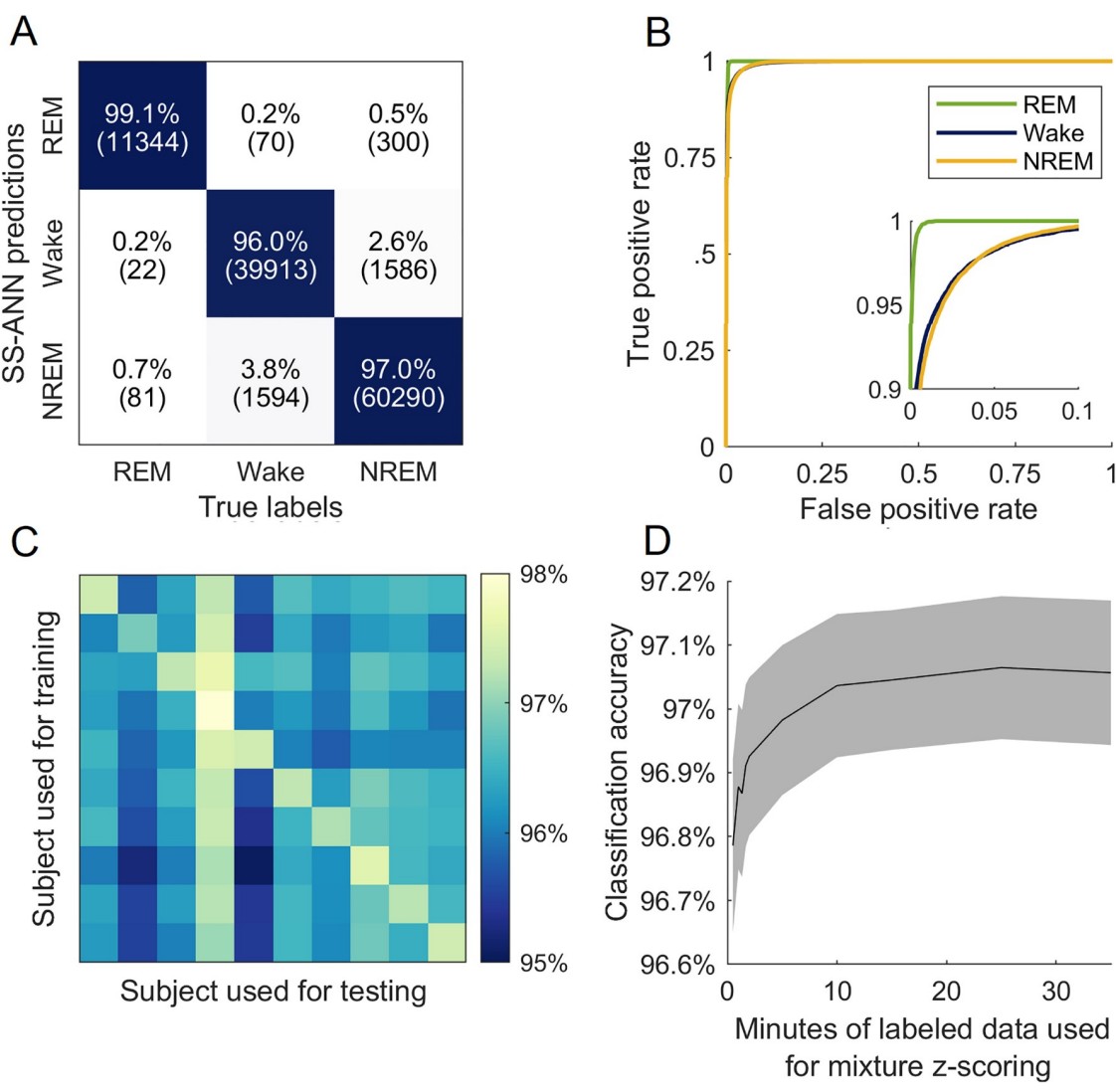

**Fig 4. Validation of SS-ANN.** A: confusion matrix for SS-ANN on held-out data. The overall accuracy was 96.8%. The number of epochs is shown in parentheses. B: receiver operating characteristic (ROC) for SS-ANN. The inset panel shows a zoomed-in view of the upper-left corner. C: generalization across subjects. SS-ANN was trained on each mouse individually and tested on all mice. D: classification accuracy as a function of the amount of labeled data used for mixture *z*-scoring. Gray shading shows SEM, n = 20 recordings.

mixture *z*-scoring is unnecessary to obtain high performance, since the animals in our cohort did not undergo any experimental manipulations and the intent of mixture *z*-scoring is to account for the possibility of observing altered class balances. Additionally, note that small sample sizes result in uncorrelated errors in the estimation of the *z*-score parameters for each feature, which have less impact on classification error than do the correlated errors caused by class balance variability.

## AccuSleep: Free, open-source software for automated sleep scoring

The benefit of mixture *z*-scoring for generalization across class balances comes with the requirement that for each subject, there must be enough labeled epochs of each class to estimate the class-conditional means and variances of each feature. In practice, labeling these

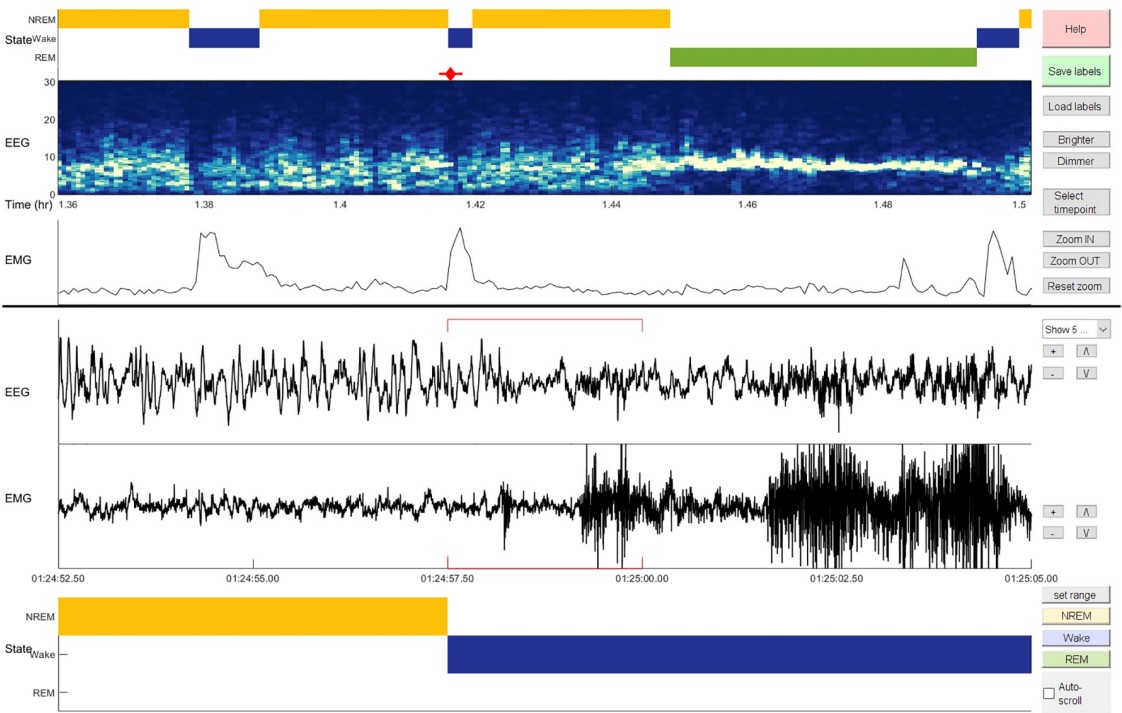

**Fig 5. AccuSleep interface for manual sleep scoring.** The lower three panels display the EEG and EMG signals as well as the sleep stage labels for epochs surrounding the currently selected epoch. The upper three panels provide context by displaying the sleep stages, EEG spectrogram, and EMG power on a longer time scale. The red line below the first panel indicates the time span of the lower three panels, and the diamond indicates the location of the currently selected epoch. For a complete description of the feature set of this software, please see the included user manual.

epochs requires a user to interact with the data in a non-trivial way. We addressed this issue by creating AccuSleep, a set of MATLAB graphical user interfaces that allow for manual scoring of EEG/EMG data (Fig 5) followed by automated scoring using SS-ANN (Fig 6). The mixture *z*-scoring parameters, once calculated for a given subject, can be used to score other recordings from the same subject. The workflow for scoring recordings is simple:

1. **Select EEG and EMG data**. In the interface shown in Fig 6, the user selects the data files, enters the sampling rate and epoch length, and sets the output location for all recordings from a given subject. Arbitrary epoch lengths can be used, provided that the neural network used for automatic scoring was trained on data scored at the same temporal resolution.

2. **Set mixture *z*-scoring parameters**. If mixture *z*-scoring parameters have already been calculated for this subject, they can be loaded. If not, the user scores a small number of epochs of each state manually using the interface shown in Fig 5. AccuSleep then calculates and saves the parameters.

3. **Load a trained copy of SS-ANN**. The user loads a network for automated classification. The trained network validated in Fig 4 is included with the software, along with MATLAB functions to retrain the network on new data.

4. **Score sleep automatically**. After validating the inputs, AccuSleep uses SS-ANN to perform automatic sleep scoring.

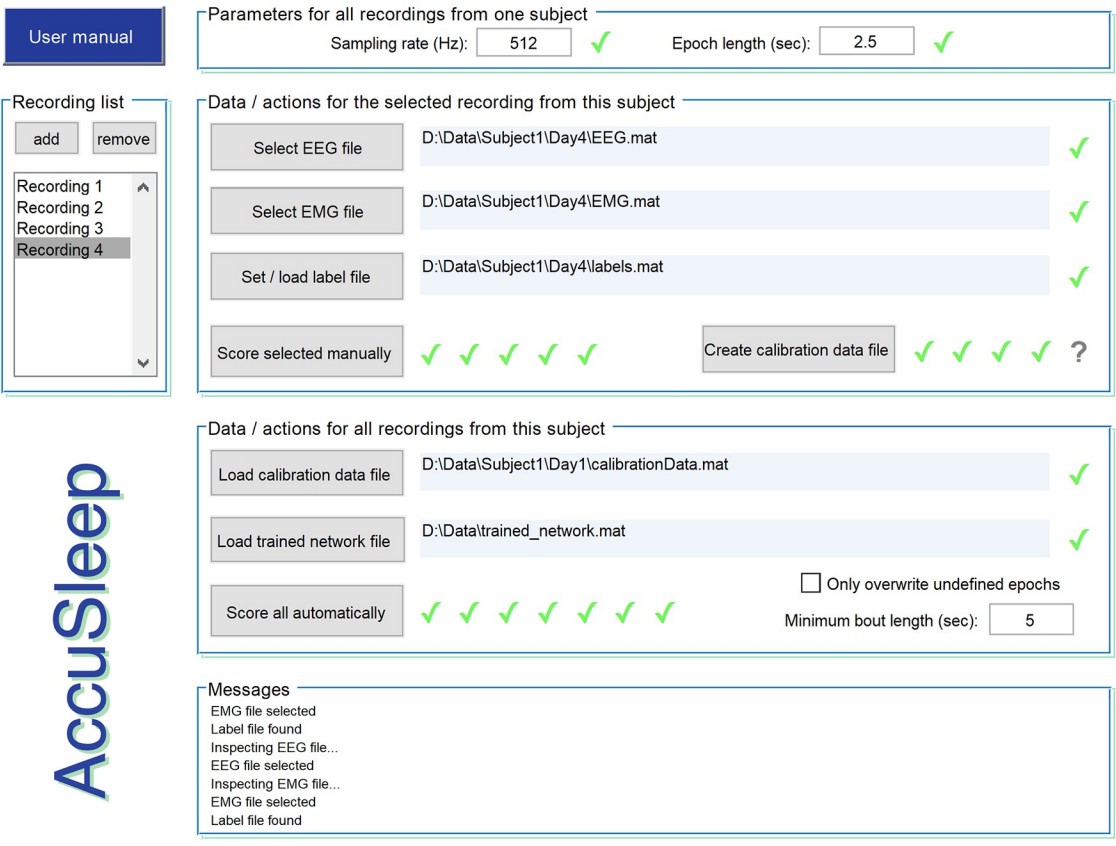

**Fig 6. AccuSleep interface for automated sleep scoring.** After a small number of epochs are manually scored (Fig 5), the software uses SS-ANN and mixture *z*-scoring to perform automated sleep scoring on all recordings from a given subject simultaneously. Green check marks indicate valid user inputs.

For an experienced user, manually scoring the number of epochs required for maximum classification accuracy (Fig 4D) requires roughly 2 minutes, far shorter than it would take to score all of the data from that subject. Thereafter, scoring of additional data from the same subject does not require any manual labeling, preserving the scaling and efficiency benefits of automated scoring.

## Discussion

Supervised machine learning is well suited to the task of sleep scoring: labeled data are plentiful, and contemporary algorithms can learn from minimally processed EEG and EMG data to achieve classification accuracy comparable to inter-rater reliability. Nevertheless, machine learning algorithms are still not widely used for sleep scoring in research. We suspect there are two reasons for this: low usability, since applying machine learning methods can require specialized knowledge or skills; and poor generalization, since variability in the EEG and EMG signals due to inter-subject and inter-laboratory differences, or distributional shift, poses a challenge to the generalization of any algorithm. Both of these issues must be addressed if machine learning is to be widely adopted for sleep scoring.

We propose mixture *z*-scoring (Eq 2) as a solution to the problem of generalization posed by distributional shift due to simultaneous nuisance and class balance variability. Standard *z*-scoring serves this purpose well when class balance variability is low, but not when class

balance variability is high, as it often is in scientific settings. In our experiments, which simulated changes in class balance such as might occur in a sleep study, we found that a classifier using standard *z*-scoring as a preprocessing step performed poorly on data from a subject with a different class balance than its training set (Fig 3B–3E). This can produce undesirable results in a research setting because the effect of a manipulation that changes the quantity of sleep or wakefulness would be poorly estimated (Fig 2F). Mixture *z*-scoring captures the two sources of variability independently, removing only the former. On both simulated (Fig 2E and 2I) and real (Fig 3B–3F) data with artificially-varied class balance, our method improves generalization and estimation of class balance. We expect that the same is true for real data with natural class balance variability—for example, recordings collected during and after a sleep deprivation protocol where the fractions of wakefulness and sleep are both different from baseline conditions.

We also introduce a new classification algorithm for rodent sleep scoring, SS-ANN (Sleep Scoring Artificial Neural Network). This convolutional neural network achieves comparable accuracy to inter-scorer agreement and to another neural network method, SPINDLE [10] (SS-ANN: 96.8%; SPINDLE: 94.8-96.2%, see Methods). This is achieved with 300x fewer parameters, at a higher temporal resolution, and without using a Hidden Markov Model to constrain transitions between sleep states. The vast majority of parameters in both networks are in the fully-connected layer that follows the convolutional component of the architecture. Thus, the difference in parameter count comes from SS-ANN's greater use of convolution and pooling. The possibility that even simpler models with fewer parameters might achieve comparable performance remains to be explored.

While mixture *z*-scoring dramatically reduces bias due to class balance variability and improves generalization, this comes at a cost: a sample of labeled data from each subject must be provided in order to capture subject-specific variability. Unsupervised, or label-free, methods for handling class balance variability would avoid this requirement, but have other costs. In [17], certain feature values are explicitly assumed to be an ordered mixture-of-Gaussians, allowing for the threshold of a linear classifier to be placed without requiring labels. Mixture *z*-scoring does not make any assumption about the shape of the class-conditional feature distributions, nor does it rely on the specific form of the downstream classifier. The two methods are equivalent in the case of mixture-of-Gaussians data and a linear classifier. Alternative unsupervised methods that do not involve distributional assumptions might include data augmentation with inputs whose nuisance and class balance variability are altered programmatically or randomly, so long as these augmentation methods are applied upstream of the preprocessing step. One advantage of our supervised method over this alternative is that it applies to any class balance and any affine nuisance variability, rather than only for ranges included in the augmentation step. Further, our method preserves any affine variability that is useful to classification, while data augmentation removes it. For example, applying data augmentation to the linear classification problem in Fig 2 would result in a classifier with poor performance. The success of threshold-based methods [3, 17] indicates that affine features, such as EMG power, are very useful for classification, suggesting that data augmentation would be harmful to performance. Finally, our results indicate that the amount of labeled data required to achieve high accuracy is on the order of minutes (Fig 4D), indicating that the time cost of the labeling step is quite small.

To preserve the speed and scalability benefits of automated scoring that includes mixture *z*-scoring, and therefore some manual labeling, as a preprocessing step, we aimed to make the labeling step as streamlined as possible. To this end, we developed AccuSleep: a free, open-source MATLAB package that provides graphical interfaces for manual and deep learning-based, automated sleep scoring (Fig 6). Within AccuSleep, polysomnographic recordings can be manually scored to provide the labeled data for mixture *z*-scoring (Fig 5). These mixture

parameters can then be used to score all recordings from the same subject automatically. Automatic classification is performed using a copy of SS-ANN trained on the data collected for these experiments (see Methods).

While SS-ANN showed good generalization across the 10 mice in our cohort (Fig 4C), it is possible that recordings collected from mice with different genotypes or differently placed electrodes would have substantially different EEG spectra from those in our dataset. To account for this possibility, we leverage one of the advantages of end-to-end learning—the fact that training new models is simple given labeled data—by including a module in AccuSleep that can train a new version of SS-ANN based on a sample of labeled data. Such a model could be provided alongside a research study to increase the replicability of its sleep scoring methodology. The time and computational resources required for this training process are minimal, owing to the small number of parameters in SS-ANN.

In summary, we developed a standardization procedure, mixture *z*-scoring, that simultaneously corrects for distributional shift due to both nuisance variability and changes in class balance. We demonstrated that our method improves the generalization of sleep scoring algorithms and provide software to enable its application in sleep research. We expect that this software, available at https://github.com/zekebarger/AccuSleep, will be useful to the research community. More broadly, we note that data standardization that goes beyond textbook *z*-scoring and accounts for class balance changes across experimental units is ubiquitous in experimental sciences (e.g., *z*-scoring of electrocorticography recordings from humans [18]). Such methods are less commonly used in a machine learning setting, where algorithms are typically formulated assuming no distributional shift and validated using test sets that have similar class balance to the training set, such as held-out data. As demonstrated here, the practice of training algorithms in this manner on one type of particularly convenient experimental subject and then applying them on another, ignoring class balance variation, can lead to incorrect scientific conclusions. This suggests that mixture *z*-scoring could improve the accuracy of machine learning algorithms across scientific domains.

## Methods

### Polysomnographic recordings

All experimental procedures were approved by the Animal Care and Use Committee at the University of California, Berkeley. Animals were housed on a 12-hour dark/12-hour light cycle (light on between 7:00 and 19:00). Adult C57BL/6 mice (10-20 weeks old) were anesthetized with 1.5%–2% isoflurane and placed in a stereotaxic frame. Body temperature was kept stable throughout the procedure using a heating pad. After asepsis, the skin was incised to expose the skull, and the overlying connective tissue was removed. For EEG and EMG recordings, a reference screw was inserted into the skull on top of the right cerebellum. EEG recordings were made from two screws on top of the left and right cortex, at anteroposterior –3.5 mm and medio-lateral +/-3 mm. Two EMG electrodes were inserted into the neck musculature. Insulated leads from the EEG and EMG electrodes were soldered to a pin header, which was secured to the skull using dental cement. All efforts were made to minimize suffering during and after surgery.

Recordings were made with the mice in their home cages placed in sound-attenuating boxes. Five 4-hour recordings were collected from each of 10 mice, and two 24-hour recordings were collected from five of those mice. For 24-hour recordings, recording started at 19:00 following 24 hours of habituation and lasted 48 hours. For four-hour recordings, recording started at 13:00 following two hours of habituation. The pin header was connected to a flexible recording cable via a mini-connector. Signals were recorded with a TDT RZ5 amplifier for the

24-hour recordings (bandpass filter, 1-750 Hz; sampling rate, 1,500 Hz) or an Intan Technologies RHD-2132 amplifier for the 4-hour recordings (bandpass filter, 1-500 Hz; sampling rate, 1,000 Hz). Sleep stages were scored manually in 2.5-second epochs by an expert scorer according to standard criteria. The complete dataset is available at https://osf.io/py5eb/.

The data used for training and testing SS-ANN in each experiment are described below. Unless otherwise specified, SS-ANN was trained using three 4-hour recordings from each of the 10 mice.

- Fig 3: tested on three programmatically rebalanced 12-hour light cycle recordings (extracted from the 24-hour recordings) from three mice

- Fig 4A and 4B: tested on two held-out 4-hour recordings from each of 10 mice

- Fig 4C, off-diagonal: trained on all five 4-hour recordings from each mouse, tested on all five 4-hour recordings from each of the other nine mice

- Fig 4C, on-diagonal: five-fold cross-validation using all five 4-hour recordings from each mouse

- Fig 4D: as for Fig 4A and 4B, but with a held-out portion of each test recording used for mixture *z*-scoring

Three mice were not used for recordings due to low signal-to-noise ratio (SNR) in the EEG and EMG signals. One mouse was excluded from our analyses because the SNR of its signals decreased before data collection was completed.

## Sleep scoring algorithm

**Data preprocessing.** EEG and EMG signals were downsampled to 128 Hz. We used the Chronux toolbox [19] to calculate a multi-taper spectrogram of the EEG signal between 0-50 Hz with a 5 second window and 2.5 second step. We downsampled by a factor of 2 between 20-50 Hz to reduce the number of parameters in the classifier. To calculate EMG activity, we bandpass filtered the EMG signal between 20-50 Hz and took the root-mean-square of the signal in each epoch. To build the complete feature set for each recording, we concatenated the EEG spectrogram with 9 copies of the EMG activity. Since the spectrogram has 176 frequency components, each recording becomes a $185 \times n$ matrix with 185 features for each of $n$ epochs.

**SS-ANN architecture.** The inputs to SS-ANN were $185 \times 13$ pixel grayscale images, representing 32.5-second periods of the standardized joint EEG/EMG spectrogram centered on each epoch. We created a basic CNN architecture using the MATLAB Statistics and Machine Learning Toolbox (MATLAB, The MathWorks, Natick, MA): 3 convolution—batch normalization—ReLU—max pooling modules, followed by a fully connected layer, softmax layer, and classification layer. The convolution layers had filter size 3 with 8, 16, and 32 filters per layer. The max pooling layer had size 2 and stride 2. The network was trained using stochastic gradient descent with momentum and a mini-batch size of 256 for 10 epochs. The learning rate was 0.015, reduced by 15% each epoch. Classes were balanced prior to training by randomly oversampling the classes with the fewest examples to reach the number of examples in the largest class. Following the classification step, sleep stages were refined by assigning bouts shorter than 5 seconds to the surrounding stage.

**Comparison between SS-ANN and SPINDLE.** Up to 20% of epochs in the recordings used for training or testing SPINDLE were scored as artifacts [10], but inter-rater agreement for artifact detection was low (in the range of 20-30%). The accuracy of SPINDLE reported in [10] was calculated only using epochs not labeled as artifacts and on which two expert raters

agreed. These criteria may remove some of the most difficult-to-classify epochs. We used the SPINDLE online service with artifact detection disabled to re-score the datasets used in that publication, obtaining accuracies of 96.2%, 95.1%, 94.8% on Cohorts A, B, and C versus the labels of Expert 1.

**Mixture $z$-scoring.** For a given set of $n + N$ observations of feature values $X$, $n$ of which have paired observations of labels $L$, and fixed choice of baseline mixture weights $w$, we perform *mixture z-scoring* to obtain standardized observations $Z_M$ as follows:

$$Z_M = \frac{X - w^\top \hat{\mu}}{\sqrt{w^\top (\hat{\sigma}^2 + s)}}$$

defining subtraction and division between vectors and scalars and squaring of vectors as the element-wise versions of their scalar equivalents, where the vectors $\hat{\mu}$, $\hat{\sigma}$, and $s$ are as below, denoting by $X_l$ the set of observations with paired label equal to $l$:

$$\hat{\mu}_l = \frac{1}{n} \sum_{x \in X_l} x \tag{3}$$

$$\hat{\sigma}_l^2 = \frac{1}{n} \sum_{x \in X_l} (x - \hat{\mu}_l)^2 \tag{4}$$

$$s_l = (\hat{\mu} - w^\top \hat{\mu}) \tag{5}$$

We found that choosing the baseline mixture weights $w$ to be close to the class prevalences in a reference dataset worked well.

This algorithm can be motivated as follows: let $\phi$ be a random variable corresponding to the feature value and $Y$ be the random variable corresponding to the class label. We can break down the distribution of the feature value, $P(\phi)$, into a mixture of the distributions conditioned on the class label, $P(\phi|Y = i)$, each with a corresponding mean $\mu_i$ and variance $\sigma_i^2$

$$w_i := P(Y = i) \tag{6}$$

$$\mathbb{E}[\phi|Y = i] := \mu_i \tag{7}$$

$$\sigma_i^2 := \mathbb{V}[\phi|Y = i] \tag{8}$$

The overall mean $\mu_G$ and variance $\sigma_G^2$ of this mixture distribution can be written in terms of the means and variances of its components as follows, writing $\mu$ and $\sigma^2$ for the vectors of class-conditional means and variances, and $w$ for the vector of class probabilities:

$$\mu_G := \mathbb{E}[\phi] = w^\top \mu \tag{9}$$

$$\sigma_G^2 := \mathbb{V}[\phi] = w^\top (\sigma^2 + s) \tag{10}$$

$$\mathbb{V}[\phi] = w^\top \sigma^2 + w^\top (\mu - w^\top \mu)^2 \tag{11}$$

$$s := (\mu - w^\top \mu)^2 \tag{12}$$

where the expression for $\mu_G$ comes from the linearity of the expectation and the expression for

$\sigma_G^2$ comes from the law of total variance:

$$\mathbb{V}[\phi] = \mathbb{E}[\mathbb{V}[\phi|Y]] + \mathbb{V}[\mathbb{E}[\phi|Y]] \tag{13}$$

The vector $s$ is analogous to a "between sum-of-squares" in an ANOVA.

In this representation, we can write the probabilistically $z$-scored version of $\phi$, which we write $\phi_Z$, as:

$$\phi_Z = \frac{\phi - \mu_G}{\sigma_G} = \frac{\phi - w^\top \mu}{\sqrt{w^\top(\sigma^2 + s)}} \tag{14}$$

The key utility of this representation is that it separates out contributions to the overall mean and variance by the means and variances of each group from contributions to the overall mean and variance from the *prevalence* of each group. Note that $w$, $\mu$, $\sigma^2$, and $s$ all typically need to be estimated from data, which results in the typical, exact form of $z$-scoring.

Now instead suppose we observe a nuisance affected version, $\tilde{\phi}$, with the same class balance $w$. We presume the nuisance variability acts to, in expectation, scale and shift the distribution of $\phi$ by scaling parameter $a$ and shift parameter $b$:

$$\tilde{\mu} := \mathbb{E}[\tilde{\phi}|Y = i] = a\mathbb{E}[\phi|Y = i] + b \tag{15}$$

$$\tilde{\sigma}^2 \phi \mathbb{V}[\tilde{\phi}|Y = i] = a^2 \mathbb{V}[\phi|Y = i] \tag{16}$$

$$\tilde{s} := (\tilde{\mu} - w^\top \tilde{\mu})^2 \tag{17}$$

The method for probabilistic $z$-scoring remains the same, with the new group means and variances substituted in:

$$\tilde{\phi}_Z = \frac{\tilde{\phi} - \tilde{\mu}_G}{\tilde{\sigma}_G} = \frac{\tilde{\phi} - w^\top \tilde{\mu}}{\sqrt{w^\top(\tilde{\sigma}^2 + \tilde{s})}} \tag{18}$$

That is, in the absence of changes in class balance, we can remove affine nuisance variability by subtracting off a weighted sum of the class-conditional means and dividing by a weighted sum of the class-conditional variances and the weighted variability of the means.

This suggests a method of $z$-scoring to remove affine nuisance variability for observations, $\hat{\phi}$, with different class balances $\hat{w}$. We compute the equivalent $\hat{\mu}$ and $\hat{\sigma}^2$, which are class-specific statistics, then plug them into Eq 14 with the weights given by $w$ instead of $\hat{w}$:

$$\hat{\phi}_{Z_m} = \frac{\hat{\phi} - w^\top \hat{\mu}}{\sqrt{w^\top(\hat{\sigma}^2 + (\hat{\mu} - w^\top \hat{\mu})^2)}} \tag{19}$$

The result, applied to a finite dataset, is the operation defined by Eq 2. In essence: we try to perform the same nuisance variability-removing $z$-scoring operation we would have done, had there not been a class balance change.

Note that in general the resulting data $Z_m$ no longer has mean 0 or standard deviation 1, since that is only true when $\hat{w} = w$.

## Acknowledgments

We would like to thank Chak Foon Tso for his assistance testing the software and Hayley Bounds for her assistance designing the neural network.

## Author Contributions

**Conceptualization:** Zeke Barger, Charles G. Frye, Danqian Liu, Yang Dan, Kristofer E. Bouchard.

**Data curation:** Zeke Barger, Danqian Liu.

**Formal analysis:** Zeke Barger, Charles G. Frye.

**Funding acquisition:** Yang Dan, Kristofer E. Bouchard.

**Investigation:** Zeke Barger, Charles G. Frye.

**Methodology:** Zeke Barger, Charles G. Frye.

**Resources:** Yang Dan, Kristofer E. Bouchard.

**Software:** Zeke Barger.

**Supervision:** Yang Dan, Kristofer E. Bouchard.

**Validation:** Danqian Liu.

**Visualization:** Zeke Barger, Charles G. Frye.

**Writing – original draft:** Zeke Barger, Charles G. Frye.

**Writing – review & editing:** Zeke Barger, Charles G. Frye, Danqian Liu, Yang Dan, Kristofer E. Bouchard.

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
