## [Decision Letter · Decision Letter 0]

6 Nov 2019

PONE-D-19-28848

Robust, automated sleep scoring by a compact neural network with distributional shift correction

PLOS ONE

Dear Mr. Barger,

Thank you for submitting your manuscript to PLOS ONE. After careful consideration, we feel that it has merit but does not fully meet PLOS ONE’s publication criteria as it currently stands. Therefore, we invite you to submit a revised version of the manuscript that addresses the points raised during the review process.

We would appreciate receiving your revised manuscript by Dec 21 2019 11:59PM. To enhance the reproducibility of your results, we recommend that if applicable you deposit your laboratory protocols in protocols.io, where a protocol can be assigned its own identifier (DOI) such that it can be cited independently in the future. For instructions see: http://journals.plos.org/plosone/s/submission-guidelines#loc-laboratory-protocols

We look forward to receiving your revised manuscript.

Kind regards,

Giorgio F Gilestro, PhD

Academic Editor

PLOS ONE

Journal Requirements:

Reviewers' comments:

Reviewer's Responses to Questions

**Comments to the Author**

1. Is the manuscript technically sound, and do the data support the conclusions?

Reviewer #1: Yes

2. Has the statistical analysis been performed appropriately and rigorously? 

Reviewer #1: Yes

3. Have the authors made all data underlying the findings in their manuscript fully available?

Reviewer #1: Yes

4. Is the manuscript presented in an intelligible fashion and written in standard English?

Reviewer #1: Yes

5. Review Comments to the Author

Reviewer #1: The manuscript by Barger and colleagues introduces a novel standardisation procedure to improve the performance of automated methods for sleep scoring, with a focus on rodent studies.

The manuscript is technically sound and has the potential to benefit the wider sleep community.

I only have some minor comments for the authors to address:

1) In order to use the propose method, manual labelling (scoring) of a few minutes of data (per subject) is needed. One potential issue is that the labeled data might not have the same class balance as the rest of the dataset. For instance, a user might label a portion of the data during which a subject is primarily awake, but thus may not represent the rest of the dataset. The authors should discuss to what extent this may be a problem, possibly quantify the impact of a mismatch between class distribution in labeled vs. unlabelled data, and introduce software-based measures to address this issue.

2) In Figure 1, it is not clear what feature of EMG activity are quantified and displayed in panel C.

3) From line 270 to line 315 the authors refer (in several instances) to Fig. 3C-D, while the correct reference is to Fig. 4C-D

4) I find Fig. 5 not clear enough. In particular, the legend should better help the reader in understanding how the bottom and top panels temporally relate to each other.

5) Does the software force to use 2.5 s epochs, or can an arbitrary epoch length be chosen?

6. PLOS authors have the option to publish the peer review history of their article (what does this mean?). If published, this will include your full peer review and any attached files.

Reviewer #1: No

---

## [Author Response · Author response to Decision Letter 0]

25 Nov 2019

We thank the reviewer for their thoughtful comments, which were quite positive. In general, we took the comments to be requests for clarification. We have modified our manuscript based on the reviewer’s comments, and our responses to the comments are below. Page and line numbers refer to the updated version of the manuscript without tracked changes.

1) In order to use the proposed method, manual labelling (scoring) of a few minutes of data (per subject) is needed. One potential issue is that the labeled data might not have the same class balance as the rest of the dataset. For instance, a user might label a portion of the data during which a subject is primarily awake, but thus may not represent the rest of the dataset. The authors should discuss to what extent this may be a problem, possibly quantify the impact of a mismatch between class distribution in labeled vs. unlabelled data, and introduce software-based measures to address this issue.

We take the reviewer’s comment to stem from concern that if the labeled sample of data used in the mixture z-scoring process does not have the same balance of classes as does the full dataset, then the estimated standardization parameters (means and variances) will be biased away from the correct values.

The labeled sample of the data is used to estimate the means and variances of observations from each class individually. In other words, K separate samples (one per class) are being collected to estimate K sets of parameters. Therefore, it is not necessary for the class balance of the overall sample to resemble that of the full dataset--it is only necessary that its K constituent samples are large enough to provide good estimates of the parameters for each class.

We have updated the text to clarify this aspect of the algorithm: the paragraph beginning on line 276, page 8 now reads “Finally, we investigated the relationship between classification accuracy and the amount of labeled data used for mixture z-scoring. As described in the Methods section, mixture z-scoring requires labeled data from each subject in order to estimate a mean and variance for each data feature within each class. Though the class balance of the labeled sample has no direct effect on the estimation of these parameters, it is important to determine how many labeled epochs are required to required to attain accurate enough parameter estimates to support classification.”

In the AccuSleep software, we do enforce a requirement that at least three epochs of each class are present in the labeled sample. Our user manual encourages users to provide larger labeled samples when possible, which should increase the accuracy of the parameter estimation.

2) In Figure 1, it is not clear what feature of EMG activity are quantified and displayed in panel C.

We follow the convention that EMG activity is defined as the root-mean-square of the EMG signal. We have updated the caption for Figure 1 (p 2) to reflect this information: it now reads, “...C: example EEG spectrograms and root-mean-square EMG activity for each sleep stage.”

3) From line 270 to line 315 the authors refer (in several instances) to Fig. 3C-D, while the correct reference is to Fig. 4C-D

We have corrected the references to Figure 4 in the text on lines 273, 287 and 321.

4) I find Fig. 5 not clear enough. In particular, the legend should better help the reader in understanding how the bottom and top panels temporally relate to each other.

We have clarified the caption for Figure 5 to better explain the relationship between the two sets of panels in the software. It now begins: “The lower three panels display the EEG and EMG signals as well as the sleep stage labels for epochs surrounding the currently selected epoch. The upper three panels provide context by displaying the sleep stages, EEG spectrogram, and EMG power on a longer time scale. The red line below the first panel indicates the time span of the lower three panels, and the diamond indicates the location of the currently selected epoch.”

5) Does the software force to use 2.5 s epochs, or can an arbitrary epoch length be chosen?

Arbitrary epoch lengths can be chosen. However, the network used for classification should be trained on data with the same epoch length as the desired output. We provide networks trained on data scored at 2.5, 4, 5 and 10 second resolution with the software. For other epoch lengths, it is possible to use the software to train a new network. We have updated the text to reflect this information: the description of the sleep scoring workflow now includes the sentence (line 308, p 9), “Arbitrary epoch lengths can be used, provided that the neural network used for automatic scoring was trained on data scored at the same temporal resolution.”

Additionally, we made the following minor changes to the text to improve clarity:

1. In the Methods section, we now indicate the genotype (C57BL/6) of the mice used in our experiments (line 427, p 11).

2. In the caption for Figure 3 (p 7), we corrected the order of the panel descriptions.

3. In the description of our neural network, SS-ANN, we changed the wording on line 212, p 6 to better reflect the fact that spectrograms of the EMG signal are not used as inputs to the network, only the log-transformed root-mean-square measurement.

4. The line styles used in Figure 2 have been switched between subject I and subject II to improve the clarity of panels D and H. The caption (p 4) has been updated to reflect this change.

5. Since submitting the manuscript, we have made several improvements to the appearance and functionality of the software we describe. Figure 6, its caption (p 9), and the description of the sleep scoring workflow on line 301, p 8 have been updated to reflect these changes.

---

## [Editor Report · Decision Letter 1]

2 Dec 2019

Robust, automated sleep scoring by a compact neural network with distributional shift correction

PONE-D-19-28848R1

Dear Dr. Barger,

We are pleased to inform you that your manuscript has been judged scientifically suitable for publication and will be formally accepted for publication once it complies with all outstanding technical requirements.

With kind regards,

Giorgio F Gilestro, PhD

Academic Editor

PLOS ONE
---

## [Editor Report · Acceptance letter]

5 Dec 2019

PONE-D-19-28848R1 

Robust, automated sleep scoring by a compact neural network with distributional shift correction 

Dear Dr. Barger:

I am pleased to inform you that your manuscript has been deemed suitable for publication in PLOS ONE. Congratulations! Your manuscript is now with our production department. 

With kind regards,

on behalf of

Dr. Giorgio F Gilestro 

Academic Editor

PLOS ONE